# OpenReview forum: "Collaborative Disagreement Resolution for Scalable Oversight"
_ICML.cc/2026/Conference — ICML 2026 regular_

### Official Review · Reviewer_SPEu · 2026-03-04

**Soundness:** 2
**Presentation:** 3
**Significance:** 2
**Originality:** 2
**Overall Recommendation:** 4
**Confidence:** 3

**Summary:**

This paper proposes Disagreement Resolution (DR), a collaborative alternative to adversarial debate for scalable oversight. The core idea is interesting: instead of requiring a weak judge to arbitrate between two fixed opposing positions, the protocol encourages consultants to identify the crux of disagreement and move toward consensus, with the judge primarily verifying the final outcome.

**Compliance With Llm Reviewing Policy:**

Affirmed.

**Final Justification:**

The authors took efforts to address my concerns.

**Key Questions For Authors:**

Please see my comments above.

**Limitations:**

I find the central idea interesting, but I do not think the current experimental design supports the paper’s strongest claims. In particular, several baseline comparisons are not budget-matched: DR and Debate are evaluated as 5-turn interactive protocols, whereas Double Consultancy and Naive Judge are implemented as single-shot baselines. In addition, the evaluation subset induces substantial consultant imbalance (roughly 30–40% vs. 60–70% accuracy), and the paper itself argues that DR is strongly consultant-pair-sensitive. As a result, it is difficult to determine how much of the reported gain is attributable to the collaborative protocol itself versus differences in inference budget and pair composition. I would be open to revising my assessment if the authors add budget-matched baselines and experiments with more balanced consultant pairs.

**Strengths And Weaknesses:**

I am not convinced the current evaluation is strong enough to support the paper’s broader claims.

My main concern is fairness of the baselines. DR and Debate are evaluated as multi-turn interactive protocols with up to 5 turns/rounds, while Double Consultancy is a one-turn baseline and Naive Judge is a single deterministic completion. In Appendix B, Double Consultancy is run with T=0 and n=1, Debate uses 5 rounds, and DR uses a different temperature setting (temperature=1 for both consultants and judge). This makes it difficult to disentangle the effect of the protocol itself from differences in interaction budget, stochasticity, and compute. A much fairer comparison would repeat or sample Double Consultancy and the single-judge baseline multiple times under a matched budget.

My second concern is that the evaluation subset induces substantial consultant imbalance: the paper explicitly notes 30–40% vs. 60–70% consultant accuracy on the natural disagreement subset. The authors describe this as arising from differing error patterns, but they do not provide experiments with more balanced consultant pairs or equally capable consultants. This matters because the paper also argues that DR is consultant-sensitive, and some of the qualitative analysis suggests that performance can be dominated by one model in the pair.

I also think the paper’s presentation overstates the generality of the results. Section 5.3 and the conclusion emphasize that DR helps most when the judge is weakest, and that the benefit shrinks or becomes negative for stronger judges. This heterogeneity should be stated much more explicitly up front. The main empirical finding is not simply that DR outperforms debate, but that it does so primarily when the judge is substantially weaker than the consultants; for stronger judges, the benefit diminishes and can become negative. The current abstract and introduction read as more general than this result warrants.

Overall, I find the central idea promising and the paper worth revising, but I do not think the current empirical evidence is yet sufficient for acceptance. I would be happy to update my assessment if the authors add (1) budget-matched/sampling-matched baselines for Double Consultancy and Naive Judge, and (2) experiments with more balanced consultant pairs or equally strong consultants. I hope the authors find the comments helpful.

---

> ### Author Rebuttal · Authors · 2026-03-31
>
> We thank the reviewer for recognizing the novelty of our work and for the constructive feedback. As requested, we conducted both (1) budget-matched baselines and (2) experiments with balanced consultant pairs. We summarize the results below.
>
> > Weakness 1: Missing budget-matched baselines for Double Consultancy and Naive Judge, making it hard to separate protocol effects from differences in budget, stochasticity, and compute.
>
> We address the two baselines separately:
>
> **Naive Judge** is included to verify capability asymmetry, not as a competing protocol. The judges’ standalone accuracies remain consistently low (below 26%), confirming that they are substantially weaker than the consultants. Since its purpose is only to measure unaided capability, a one-turn baseline is appropriate.
>
> **Double Consultancy**: We reran experiments with budget-matched settings (`temperature=1`, 5 reasoning turns), matching Debate and DR:
>
> | Pair | Judge | GPQA Debate | GPQA DC | GPQA DR | SuperGPQA Debate | SuperGPQA DC | SuperGPQA DR | HLE Debate | HLE DC | HLE DR |
> |---|---|---:|---:|---:|---:|---:|---:|---:|---:|---:|
> | GPT-4o / Claude | gemma-3-4b | 50.0% | 47.6% → 52.2% | **65.9%** | 52.0% | 60.8% → 61.4% | **64.7%** | 44.3% | **59.1%** → 56.8% | 37.5% |
> | GPT-4o / Claude | gpt-4o-mini | 63.9% | 54.0% → 60.9% | **67.5%** | **68.3%** | 52.3% → 66.7% | 64.7% | 34.7% | 52.3% → **58.0%** | 38.6% |
> | GLM-4.6 / Kimi K2 | gpt-4o-mini | 69.2% | 59.4% → 66.7% | **77.2%** | 41.7% | 51.4% → 52.8% | **66.2%** | 32.6% | 46.2% → 51.2% | **58.8%** |
> | GLM-4.6 / Kimi K2 | gemma-3-4b | 46.8% | 60.4% → 45.2% | **74.3%** | 46.2% | 62.5% → 56.9% | **66.7%** | 41.3% | 40.0% → 55.0% | **62.5%** |
>
> Notation: `DC` denotes Double Consultancy (`old → new`). Bold marks the best result among Debate, DC, and DR for each dataset. Overall: DR > 5-turn Double Consultancy > Debate.
>
> These results show that increasing Double Consultancy’s budget does not qualitatively change the findings: DR still outperforms both Debate and budget-matched Double Consultancy. We will include these results in the revision.
>
> > Weakness 2: Missing experiments with more balanced or equally strong consultant pairs, since the paper also argues that DR is consultant-sensitive and may be dominated by one model in the pair.
>
> As requested, we reran experiments on a rebalanced dataset where we randomly downsampled examples so that the two consultants have an approximately 50–50 split in initial correctness:
>
> | Consultant Pair | Judge Model | GPQA | SuperGPQA | HLE |
> |---|---|---|---|---|
> | GPT-4o vs Claude Sonnet 4 | gemma-3-4b | 43.5% → **68.0%** (+24.5%) | 41.5% → **59.6%** (+18.1%) | **36.7%** → 34.4% (-2.3%) |
> | GPT-4o vs Claude Sonnet 4 | gpt-4o-mini | 56.5% → **66.0%** (+9.5%) | 51.6% → **60.6%** (+9.0%) | 24.2% → **35.9%** (+11.7%) |
> | GLM-4.6 vs Kimi K2 Thinking | gpt-4o-mini | 61.8% → **76.5%** (+14.7%) | 30.4% → **62.2%** (+31.8%) | 39.5% → **51.6%** (+12.1%) |
> | GLM-4.6 vs Kimi K2 Thinking | gemma-3-4b | 43.1% → **75.0%** (+31.9%) | 39.1% → **66.7%** (+27.6%) | 41.1% → **49.2%** (+8.1%) |
>
> Notation: $a\% \rightarrow b\%$ denotes Debate $\rightarrow$ DR, and the value in parentheses is $\Delta = \text{DR} - \text{Debate}$.
>
> As shown above, DR continues to outperform Debate in nearly all settings even after rebalancing the consultant pair. This suggests that our main findings are robust under both the original imbalanced sampling and a substantially more balanced setting.
>
> Compared with Table 2 in the paper, the overall patterns remain similar, and differences between balanced and imbalanced sampling are limited. We believe this is because DR depends more on consultant calibration than on surface-level strength on a finite sample set, consistent with Theorem 3, Lemma 4, and Section 5.2. Thus, although consultant imbalance may affect the size of DR’s gains, DR’s advantage over Debate does not appear to be merely an artifact of one consultant being stronger.
>
> > Weakness 3: Overstating the generality of the results.
>
> We agree that the current framing should more clearly reflect the conditional nature of our empirical findings, and we will revise the paper accordingly. In particular, we will make this heterogeneity explicit in the abstract and introduction. We will revise the opening to emphasize that our main empirical result is not that DR uniformly outperforms Debate, but that DR is most beneficial when the judge is much weaker than the consultants, precisely the regime in which Debate can break down under capability asymmetry.
>
> More broadly, our goal is not to claim that DR is a universally superior replacement for Debate, but to introduce the protocol and study when it helps, when it does not, and why.
>
> ---
> We hope these additional experiments address the reviewer’s main concerns. Since these were the conditions the reviewer highlighted for potentially updating their assessment, we kindly ask them to reconsider their score in light of the new results.

---

> > ### Author Rebuttal · Reviewer_SPEu · 2026-04-03
> >
> > Please update the paper with the changes.

---

> > > ### Author Response · Authors · 2026-04-03
> > >
> > > Dear Reviewer SPEu,
> > >
> > > Thank you for your follow-up. Due to conference policy, we cannot revise the PDF during the discussion period; however, all requested changes, including the new experimental data, are detailed in our rebuttal and will be integrated into the final version.
> > >
> > > As requested, we have conducted the budget-matched baselines and balanced-pair experiments. These results consistently support our original findings, and we hope this comprehensive additional evidence addresses the concerns you raised.
> > >
> > > We look forward to your follow-up questions and are happy to discuss them. If our responses have resolved your concerns, we would be very grateful if you would consider re-evaluating our submission and updating the score.

---

### Official Review · Reviewer_WxCU · 2026-03-12

**Soundness:** 3
**Presentation:** 2
**Significance:** 2
**Originality:** 3
**Overall Recommendation:** 4
**Confidence:** 3

**Summary:**

The paper develops math formalisms and experiments to show that a weaker agent makes better judgements when mediating between two collaborative stronger agents seeking common truth than when presiding over a debate between two antagonist stronger agents seeking to win. This advocates for a weaker mediator mediating between two stronger agents as a superior method of scalable oversight (e.g. humans over increasing AGI/ASI AI models)

**Compliance With Llm Reviewing Policy:**

Affirmed.

**Key Questions For Authors:**

1. Why invent new term "disagreement resolution" instead of the more common, well-known terms like mediation, arbitration, collaboration, etc.
2. Isn't there more recent directly relevant research using MAS under terms in #1 (some even comparison/derivatives of 'debate'?)

**Limitations:**

Not quite. Isn't 'scaleable observability" a misnomer? It seems the less capable judge/mediator/human is largely reduced to mechanically reacting to the very limited input/output of the intellectual black box they observe in the superagent collaboration system (esp when they agree).

**Strengths And Weaknesses:**

SOUNDNESS: Within the limited, and somewhat intuitive, confines of the experiments, the paper was largely sound.
PRESENTATION: The introductory text is a bit repetitive and information sparse whereas the Methodology section is poorly formatted and somewhat excessively theoretic for claims that are evasively conceptional. While the approximation Theorem 2 and Lemma 2 bring concise clarity, Theorems 1-3 are seem formalizations of self-evident intuitions rather than deriving new insights. For example, Theorem 1 says to get guaranteed value from a consultant, the consultant has to be better than the judge which is clear once 'capability' is defined as monotone with baseline accuracy.
SIGNIFICANCE: It's an interesting, if obvious finding that two collaborating super agents seeking a common middle ground would produce better consul than two super agents trying to win a debate by any means at their disposal. The paper undermines it's main stated application somewhat as a mechanism to provide scalable oversight for weaker humans controlling stronger AI agents. The paper shows that on topics beyond the weaker judge/mediator reasoning horizon, they largely just agree with the decision of more capable agents when they are in agreement. This is not oversight, merely a different type of blackbox decision-making.


ORIGINALITY:

---

> ### Author Rebuttal · Authors · 2026-03-31
>
> Thank you for acknowledging the soundness of our work. We address your concerns and questions as follows:
>
> > Weakness 1: The introductory text is a bit repetitive and information sparse whereas the Methodology section is poorly formatted and somewhat excessively theoretic for claims that are evasively conceptional. Theorems 1 and 3 seem formalizations of self-evident intuitions rather than deriving new insights.
>
> Thank you for this feedback. We will revise the introduction to reduce repetition and improve clarity, and refine the methodology section for better presentation in the camera-ready version. We include Theorems 1 and 3 to formalize and clarify the conditions under which Disagreement Resolution (DR) is necessary and effective. Specifically, Theorem 1 shows that when there is no capability asymmetry between the judge and consultants, introducing a hierarchical protocol (such as DR and Debate) provides no benefit. This result helps precisely define the regime in which scalable oversight protocols are meaningful. Theorem 3 further characterizes how the degree of capability asymmetry impacts performance, showing that as this asymmetry increases, DR has a greater advantage over Debate. This provides a concrete, testable prediction about when DR should outperform existing approaches.
>
>
> > Weakness 2: The paper undermines its main stated application somewhat as a mechanism to provide scalable oversight for weaker humans controlling stronger AI agents. DR is merely a different type of blackbox decision-making rather than oversight.
>
> According to Bowman et al. (2022), *scalable oversight* is defined as "the problem of supervising systems that potentially outperform us on most skills relevant to the task at hand". Prior work [Khan et al., 2024; Buhl et al., 2025] proposes Debate as a scalable oversight protocol to mitigate information asymmetry, but it can fail under large capability asymmetry when judges cannot reliably distinguish competing arguments, which motivates our Disagreement Resolution (DR) protocol that shifts from competition to structured, collaborative verification.
>
> We agree that in DR, the judge observes the final transcript and follows consultant consensus, which may appear like black-box adoption. However, the key difference is *what* the judge observes. In Debate, the judge receives two competing narratives with no resolution and must determine a winner. In DR, the transcript contains structured crux identification, explicit conflict enumeration, and stepwise belief updates (see Appendix D for a concrete example), which narrows the disagreement to a specific verifiable point. This makes the reasoning process more auditable even when the judge ultimately follows the consensus.
>
> Furthermore, when consultants disagree at termination, the judge does actively evaluate both positions based on logical coherence and internal consistency (Appendix D, Final Judge Prompt). So DR is not merely a different decision rule: it restructures the interaction to produce a transcript that is more interpretable for a weak judge than Debate's adversarial output.
>
> > Question 1: Why invent new term "disagreement resolution" instead of the more common, well-known terms like mediation, arbitration, collaboration, etc.
>
> We appreciate the suggestion. These are all reasonable alternatives, and we agree there is overlap with concepts like mediation and collaboration. We chose "Disagreement Resolution" primarily because it highlights what we see as the core mechanism: explicitly identifying *where* models disagree (the crux) and resolving that specific point. Terms like "mediation" or "arbitration" emphasize the judge's role, whereas DR's advantage comes from the consultants themselves isolating their divergence. That said, we are open to adjusting the terminology if it improves clarity, and we will consider the reviewer's suggestions in the revision.
>
>
> > Question 2: Isn't there more recent directly relevant research using MAS similar to DR (some even comparison/derivatives of 'debate'?)
>
> Yes, we do discuss several relevant MAS works in our Related Work section, including Prover-Estimator Debate [Brown-Cohen et al., 2025] and collaborative multi-agent debate for error detection [Chen et al., 2025]. Similar to our response to Reviewer Yjtb regarding [Du et al., 2024] and [Liang et al., 2024], these works use multi-agent interaction to improve model performance, whereas our focus is scalable oversight under capability asymmetry. But we will update our Related Work section to include these additional references and make the distinctions more comprehensive.
>
> ---
> We thank you for your time and thoughtful feedback. We hope our responses have addressed the raised concerns, and we would be grateful if the reviewer would consider updating their assessment. We are happy to provide further clarification if helpful.

---

> > ### Author Rebuttal · Reviewer_WxCU · 2026-04-04
> >
> > The smaller issues like nomenclature and justification of board line tautological formalism are somewhat addressed. However, the main critique remains which is also the main raison d'etre of this paper remains that this DR methodology is not really scalable or explainable but rather really just a black box that humans do not (given the 99.5% adoption of consensus reached in 96% of cases where they converge despite the exception pointed to in the Appendix) question in depth and certainly could not at scale. These figures directly undermine the claim human judges "actively evaluates both positions based on logical coherence and internal consistency". Unexplored is the very likely concept that lesser able human judges often would not understand or be easily mislead even when exploring traces between the DR agents. This paper needs to dramatically scale back and qualify this claim of scalable oversight.

---

> > > ### Author Response · Authors · 2026-04-05
> > >
> > > Dear Reviewer WxCU,
> > >
> > > Thank you for your continued engagement and for pushing us to sharpen our claims. This is exactly the kind of feedback that strengthens the paper.
> > >
> > > On the high adoption rate: given that consultants are frontier-level models that substantially outperform the judge, following their consensus is a rational decision under capability asymmetry. **This high adoption rate is expected and by design.** Even accounting for agreement traps, DR achieves 62.1% judge accuracy vs. 49.2% for Debate (significant in 8/12 conditions), because DR trades Debate's guaranteed adjudication burden (near-random for weak judges, e.g., gemma-3-4b at ~47% in Table 2) for a protocol where the majority outcome is correct consensus.
> > >
> > > Although the high adoption rate may mask this, the judge does retain an active role by design: it reviews the full interaction trace (crux identification, conflict resolution, confidence statements; Appendix C.3-C.4), can override consensus, and makes the final call when consultants disagree at termination. Moreover, our calibration analysis (Section 5.2) suggests that low-confidence consensus can serve as a signal to flag likely agreement traps, since well-calibrated models express higher confidence when converging on the correct answer.
> > >
> > > That said, we agree our framing overstates the judge's active role in the consensus case. We will revise to state more precisely that DR improves the *quality of what the judge observes*, not that the judge independently verifies every conclusion.
> > >
> > > We are grateful for your feedback and support.

---

### Official Review · Reviewer_wyFA · 2026-03-12

**Soundness:** 2
**Presentation:** 2
**Significance:** 3
**Originality:** 3
**Overall Recommendation:** 4
**Confidence:** 4

**Summary:**

The paper studies scalable oversight in settings where a weak judge must evaluate answers proposed by stronger AI systems. Instead of using adversarial debate, where two agents defend fixed positions, the authors propose a collaborative protocol called Disagreement Resolution. In this setup, two consultants start with different answers, exchange arguments over several rounds, and are allowed to revise their positions based on the identified crux of their disagreement. The goal is for the consultants to either converge to a shared answer or isolate the key point of disagreement, after which the judge selects the final answer based on the transcript. The method is evaluated on expert-level QA benchmarks, where the authors report that collaborative resolution generally improves judging accuracy compared to standard debate, particularly when the judge is substantially weaker than the consultants.

**Compliance With Llm Reviewing Policy:**

Affirmed.

**Final Justification:**

The authors have addressed several of my concerns in a thoughtful and constructive manner. In particular, the clarification of the trade-offs between Disagreement Resolution (DR) and Debate, as well as the quantitative analysis of outcomes (e.g., truth consensus vs. agreement traps), significantly improved my understanding of the method and its empirical behavior.

However, some limitations remain. In particular, the protocol relies on the assumption that at least one consultant holds the correct answer, leaving the “both-wrong” scenario unaddressed. Additionally, while agreement traps are now clearly acknowledged and quantified, they are not fully mitigated, which raises questions about robustness in practice. These aspects weaken the generality of the claims, especially in settings with correlated errors or less reliable consultants.

Despite these concerns, I find the core idea compelling and the empirical results sufficiently strong to warrant acceptance. I therefore update my recommendation to weak accept, with the suggestion that future work more directly address these failure modes.

**Key Questions For Authors:**

See questions and concerns in the weaknesses section.

**Limitations:**

I think the authors adequately address limitations.

**Strengths And Weaknesses:**

Strengths:
- The paper proposes a new protocol for scalable oversight that shifts multi-agent reasoning from adversarial debate toward collaborative disagreement resolution, which is an interesting and conceptually novel direction.

- I like that the paper is transparent about disagreement resolutions outcomes and the paper does a good job at presenting an alternative to classic debate rather than trying to claim that DR is superior in all aspects.

- The approach is well motivated by the observation that debate incentivises persuasion rather than truth-seeking, and the proposed mechanism attempts to mitigate this by encouraging agents to identify and resolve the key point of disagreement.

- The work addresses an important and timely problem, how weaker human or model judges can supervise stronger AI systems and evaluates the proposed protocol on challenging expert-level benchmarks.

Weaknesses:
- The methodology section is overly formal and fine-grained making the actual algorithm hard to parse. The write up should in my opinion go along the lines of Figure 1 which is more intuitive. Another approach would be to separate formalisation of the different components(actors) and the actual algorithm.
- I have similar concerns with regards to the results section. While the results are written up nicely, some of the tables  and figures themselves are a bit overloaded. For instance, I don't think we need baseline results in Table 2. Table 4 is also quite hard to interpret.
- The disagreement resolution appears to rely on the consultants resolving their disagreement toward a truth-seeking direction. However, the consultants could also agree on wrong assumptions proposed by either of the consultants. How is this addressed? Or is correctness only a secondary metric here?
- In the same sense, an advantage I see of the Debate method over the Disagreement Resolution method is that a wrong assumption or argument gets attacked by the adversary. Would this also happen with Disagreement Resolution if for instance both consultants are wrong?

Overall this paper is quite thorough and the results generally supports the proposed protocol. However, I am not fully convinced that DR simplifies oversight in practice, as it shifts the burden from a strong judge to agreement between two consultants, which assumes they are both well aligned and not subject to correlated errors. In some settings, investing in a stronger judge may be more reliable and perhaps the only course of action. However, the authors are in my opinion quite transparent about the approach which I appreciate.

The main issue with the paper is clarity: improving the exposition of the methodology would likely make the contribution easier to assess and could bring the paper above the acceptance threshold.

---

> ### Author Rebuttal · Authors · 2026-03-31
>
> Thank you for recognizing the comprehensiveness of our proposed protocol and the thoroughness of our empirical evaluation. We provide the following clarifications in response to your concerns and suggestions:
>
> > Weakness 1: Improving clarity of mathematical formulation and table / figure interpretation.
>
> Thank you for raising this concern. We will simplify the presentation of the mathematical formulation and incorporate a definition tree diagram for Debate and Disagreement Resolution (DR) in the camera-ready version.
>
> > Weakness 2: Consultants might converge on incorrect assumptions rather than truth, questioning whether the protocol lacks a mechanism to prevent "agreement traps" or if factual correctness is a secondary priority.
>
> We agree that agreement traps are a real failure mode. However, it is important to compare against the alternative. In Debate, the exit mode is *always* disagreement collapse, and a weak judge must choose between two competing arguments (near-random for weak judges). In DR, our exit mode analysis (Figure 3) shows that truth consensus occurs in 59-66% of cases, which the judge adopts correctly with 99.5%+ accuracy (Table 3). Agreement traps occur in 30-41% of cases, but this is the cost of enabling truth consensus, and the net trade-off favors DR.
>
> On mitigation: we show both theoretically (Theorem 3, Lemma 4) and empirically (Section 5.2, Figure 2) that improving consultant calibration reduces agreement trap frequency. The root cause traces to specific per-turn behaviors like overthinking and stubbornness (Table 4), pointing to calibration improvement as a concrete mitigation path. So while DR does not eliminate agreement traps, it replaces Debate's guaranteed failure mode (100% disagreement collapse for weak judges) with a distribution where the majority outcome is correct consensus.
>
> > Weakness 3: Unlike the Debate method, where adversaries actively challenge each other, DR may lack a robust mechanism to expose and correct scenarios where both consultants hold wrong answers.
>
> We respectfully disagree that Debate has a meaningful advantage in the both-wrong case for weak judges. While adversarial dynamics may expose flaws in wrong reasoning, the weak judge still faces two wrong arguments and must pick a winner. Under the capability asymmetry we study (Theorem 3), the judge cannot reliably distinguish which wrong answer is "less wrong," so Debate's adversarial structure does not translate into better accuracy in the both-wrong scenario.
>
> Our protocol restricts to instances where at least one consultant holds the correct answer and the consultants initially disagree (Section 3, footnote 2). The motivation is precisely the capability asymmetry that our work addresses: when both consultants are wrong, no protocol -- adversarial or collaborative -- gives a weak judge a realistic path to recovering the correct answer. We note that standard debate protocols [Irving et al., 2018; Michael et al., 2023; Khan et al., 2024] equally exclude this scenario, by *assigning* one debater to defend the correct answer and the other the incorrect answer. Our setup is in fact less restrictive: rather than force-assigning roles (which requires knowing the correct answer in advance), we rely on a natural disagreement filter where at least one consultant holds the correct answer. Empirically, both-wrong cases are relatively rare among our frontier-capable consultant pairs. We acknowledge that handling both-wrong cases is an important direction for future work, but we believe the more practical and impactful goal is first improving oversight for the common case where at least one frontier-capable consultant already holds the correct answer -- surfacing that existing correct answer through a better protocol yields immediate, concrete gains, whereas recovering from cases where models more capable than the judge are both wrong is a fundamentally harder problem.
>
> > Weakness 4: DR shifts the burden from the judge to consultant agreement and assumes a lack of correlated errors, but investing in a stronger judge might be more reliable in certain settings.
>
> We agree that improving judge capability is an important direction for scalable oversight. However, our work focuses on the regime of capability asymmetry between the judge and consultants (e.g., a lay human judge versus PhD-level expert models). In such settings, simply strengthening the judge may not always be feasible or cost-effective. Our goal is to show that, under this asymmetry, encouraging collaborative, truth-seeking interactions between consultants in DR can mitigate the performance collapse observed in competitive Debate, and thereby improve the final decision accuracy of weak judges.
>
> ---
> We appreciate your careful reading and valuable suggestions. We hope our responses have resolved your concerns, and we would be grateful if the reviewer would consider updating their assessment. We welcome any further questions.

---

> > ### Author Rebuttal · Reviewer_wyFA · 2026-04-04
> >
> > The authors have addressed several of my concerns in a thoughtful and constructive manner. In particular, the clarification of the trade-offs between Disagreement Resolution (DR) and Debate, as well as the quantitative analysis of outcomes (e.g., truth consensus vs. agreement traps), significantly improved my understanding of the method and its empirical behavior.
> >
> > However, some limitations remain. In particular, the protocol relies on the assumption that at least one consultant holds the correct answer, leaving the “both-wrong” scenario unaddressed. Additionally, while agreement traps are now clearly acknowledged and quantified, they are not fully mitigated, which raises questions about robustness in practice. These aspects weaken the generality of the claims, especially in settings with correlated errors or less reliable consultants.
> >
> > Despite these concerns, I find the core idea compelling and the empirical results sufficiently strong to warrant acceptance. I therefore update my recommendation to weak accept, with the suggestion that future work more directly address these failure modes.

---

> > > ### Author Response · Authors · 2026-04-05
> > >
> > > Dear Reviewer wyFA,
> > >
> > > Thank you for your continued engagement with our rebuttal. Your feedback on the trade-offs between DR and Debate, and your questions about truth consensus vs. agreement traps, genuinely helped us improve the paper.
> > >
> > > On your remaining concerns about the both-wrong setting and agreement traps: we agree these are important open questions. Our current paper does provide initial analysis showing that **improved calibration mitigates agreement traps** (Theorem 3, Lemma 4, Section 5.2), but we will make this much more salient in the updated draft. We see addressing both-wrong scenarios and developing stronger mitigations as exciting next steps that build directly on the framework we introduce here.
> > >
> > > We are grateful for your feedback and support!

---

### Official Review · Reviewer_Yjtb · 2026-03-13

**Soundness:** 2
**Presentation:** 3
**Significance:** 2
**Originality:** 2
**Overall Recommendation:** 3
**Confidence:** 4

**Summary:**

This paper proposes Disagreement Resolution (DR), a scalable oversight protocol that replaces adversarial debate with collaborative truth-seeking. In standard debate, two AI consultants defend fixed positions while a weaker judge adjudicates; in DR, consultants iteratively identify the crux of their disagreement, update their beliefs, and may adopt the opposing view, potentially converging on a consensus that the judge then verifies. The paper provides a theoretical framework with three theorems establishing that capability asymmetry is necessary for consultancy to help, that DR performance is primarily driven by consultant calibration rather than judge capability, and that DR outperforms debate for sufficiently weak judges. Experiments on three expert-level benchmarks (GPQA, SuperGPQA, HLE-MC) with two consultant pairings and multiple judge models show DR achieving 62.1% average judge accuracy compared to 49.2% for debate, with advantages most pronounced when the judge is weakest. The paper also introduces a taxonomy of consultant actions and exit modes, using Brier score-based calibration analysis to explain when and why DR succeeds or fails. The analysis reveals that agreement traps remain an unresolved failure mode of the protocol.

**Compliance With Llm Reviewing Policy:**

Affirmed.

**Final Justification:**

I remain slightly leaning toward rejection, as I do not feel the authors’ rebuttal fully resolves my concerns about the fundamental weaknesses of DR, particularly around agreement traps and underlying assumptions, despite their thoughtful responses. I would value hearing other reviewers’ perspectives on whether these limitations are sufficiently addressed or acceptable given the paper’s contributions.

**Key Questions For Authors:**

The sample sizes after disagreement filtering are relatively small (72–153 instances per condition). Given that several performance margins are narrow (e.g., +3.6% or −3.6%), have you conducted statistical significance tests or reported confidence intervals?

The DR protocol assumes that models can reliably perform cognitively demanding multi-turn operations: identifying cruxes, genuinely updating beliefs based on opposing evidence, and accurately self-reporting confidence. In practice, in my opinoin many models struggle with multi-turn consistency. Your own results show GPT-4o exhibiting high volatility regardless of answer correctness, and in the Appendix D example, a consultant switches to the wrong answer before recovering. How robust is DR to models that fail to perform principled belief updating, and to what extent are the results dependent on specific multi-turn reasoning capabilities that current models possess unevenly?

**Limitations:**

yes

**Strengths And Weaknesses:**

Strengths:

The taxonomy of consultant actions (Persistence, Recovery, Overthinking, Stubbornness), exit modes (Truth Consensus, Agreement Trap, Disagreement Collapse), and the calibration-based analysis using Brier scores provides a vocabulary and measurement framework that future work can adopt regardless of the specific protocol being studied.

Weaknesses:

The methodology shares significant overlap with several relevant works that are not discussed in the related work section:

- Goel et al. (2025), "Great Models Think Alike and this Undermines AI Oversight"
- Liu et al. (2025), "Synthetic Socratic Debates: Examining Persona Effects on Moral Decision and Persuasion Dynamics"
- Du et al. (2024), "Improving Factuality and Reasoning in Language Models through Multiagent Debate"
- Liang et al. (2024), "Encouraging Divergent Thinking in Large Language Models through Multi-Agent Debate"

Please discuss these papers. In related work the paper mentions the debate hacking phenomenon where a sophisticated liar can potentially out-argue a truth-teller, especially when the judge lacks domain expertise. However, the paper directly assumes that consultants always produce reliable reasoning (line 95-99). Not sure why this assumption makes sense here.

The paper does propose DR and claim it outperforms debate. However, the paper's own analysis reveals that agreement traps, where both consultants converge on the wrong answer and the judge unconditionally adopts it, represent an irrecoverable failure mode unique to DR, yet the paper does not propose any detection mechanism or improvements. Is this issue because of your prompt is flawed?

All three benchmarks are multiple-choice. This is a meaningful limitation because multiple-choice constrains the answer space to a small finite set (Ω(x)), which makes the judge's task artificially easier (pick from 4–5 options) and makes consensus verification trivial ("do both consultants say B?"). In real-world scalable oversight, evaluating open-ended code, proofs, medical diagnoses, the answer space is vast, consensus is harder to define, and verifying agreement is itself nontrivial.

---

> ### Author Rebuttal · Authors · 2026-03-31
>
> We sincerely thank the reviewer for recognizing our contribution in proposing a scalable oversight framework for future work. Below we address the main concerns with clarifications and additional analyses.
>
> > Weakness 1: Related work on “debate hacking” seems to conflict with the assumption of reliable consultants.
>
> Thank you for pointing out these papers. We will clarify this comparison in the revision.
>
> **Goel et al. (2025)** and **Liu et al. (2025)** study model-level biases (e.g., affinity bias, correlated errors, persona effects) that can affect both Debate and DR. Our contribution is instead at the **protocol level**: a new interaction mechanism with structural advantages despite such biases. When such biases do affect protocol behavior, especially correlated errors leading to agreement traps, we already analyze this through calibration (Section 5.2, Figure 2).
>
> **Du et al. (2024)** and **Liang et al. (2024)** use multi-agent interaction to improve **model performance**. By contrast, our focus is **scalable oversight**: improving a weaker judge’s accuracy under capability asymmetry through collaborative disagreement resolution.
>
> We also clarify that “reliable reasoning” (lines 95–99) is an idealization in the theory, while its violations are studied empirically through agreement traps (Section 5.2, Figure 2, Table 4).
>
> > Weakness 2: No detection or mitigation for agreement traps, a unique DR failure mode.
>
> In Debate, the exit mode is always **disagreement collapse** (Definition 3.3): the weak judge must resolve competing arguments every time, often near random for weak judges. In DR, despite agreement traps, **truth consensus occurs in 59–66% of cases** (Figure 3), and judges adopt it with **99.5%+ accuracy** (Table 3). Thus, agreement traps are a real cost of consensus, but the overall trade-off still favors DR.
>
> On mitigation, both theory (Theorem 3, Lemma 4) and experiments (Section 5.2, Figure 2) suggest that better consultant calibration reduces agreement traps. Table 4 further links traps to misleading behaviors such as **overthinking** and **stubbornness**, pointing to a concrete mitigation direction.
>
> > Weakness 3: Benchmarks are limited to multiple choice.
>
> We acknowledge this limitation. We follow common scalable-oversight practice [Khan et al., 2024; Kenton et al., 2024], which largely relies on multiple-choice QA, and intentionally choose **difficult, high-quality benchmarks** where capability asymmetry is most salient. Accordingly, we use GPQA, SuperGPQA, and HLE.
>
> > Question 1: Given that several performance margins are narrow (e.g., +3.6% or −3.6%), have you conducted statistical significance tests or reported confidence intervals?
>
> We computed bootstrapped confidence intervals for the judge-accuracy difference $\Delta=\text{DR}-\text{Debate}$ using $N=10{,}000$ resamples at $\alpha=0.05$.
>
> | Dataset | Judge | GPT-4o vs Claude | GLM vs Kimi |
> |---|---|---|---|
> | GPQA | 4o mini | [-5.95%, 13.1%] | [3.96%, 21.8%]* |
> |  | gemma | [5.56%, 27.4%]* | [20.0%, 42.9%]* |
> | SuperGPQA | 4o mini | [-12.4%, 4.9%] | [16.9%, 40.8%]* |
> |  | gemma | [3.59%, 21.2%]* | [20.4%, 44.4%]* |
> | HLE | 4o mini | [-7.95%, 14.8%] | [9.38%, 38.1%]* |
> |  | gemma | [-18.8%, 6.25%] | [4.43%, 32.9%]* |
>
> DR improves significantly over Debate in **8/12 settings**, reinforcing two findings from Section 5: DR yields larger gains for **weaker judges** and is **consultant-pair sensitive**, with especially strong gains for GLM-4.6 vs Kimi K2 Thinking.
>
> > Question 2: How robust is DR to models that fail to perform principled belief updating, and to what extent are the results dependent on specific multi-turn reasoning capabilities?
>
> This is an important question. DR is consultant-sensitive (Theorems 2–3, Lemma 4): performance depends more on consultant calibration than judge capability.
>
> First, DR shows **monotonic convergence** across all three datasets. For GPT-4o vs Claude, truth consensus rises from **37.3% to 58.6%** from turn 1 to turn 5, while disagreement collapse drops from **45.5% to 0.5%**. For GLM-4.6 vs Kimi K2, truth consensus rises from **54.8% to 65.9%**, while disagreement collapse drops from **26.2% to 4.0%**. This suggests genuine convergence rather than random switching.
>
> Second, DR remains robust to **heterogeneous updating styles**. Table 4 shows that GPT-4o is volatile, Kimi K2 strongly maintains conviction, and Claude is occasionally rigid. Despite this, DR still outperforms Debate for both consultant pairs (Table 2). We believe the structured crux-identification mechanism helps channel these diverse behaviors into productive convergence.
>
> ---
> We appreciate the reviewer’s thoughtful feedback and hope these clarifications and additional analyses help address the concerns. We would be grateful if the reviewer would reconsider their assessment in light of these responses.

---

> > ### Author Rebuttal · Reviewer_Yjtb · 2026-04-03
> >
> > Thanks for the response. My current concerns remain partially addressed. The assumption of reliable, cooperative consultants remains central to the method despite the paper’s motivation involving potentially deceptive models. Agreement traps are acknowledged but still lack a concrete detection or mitigation mechanism, which is a issue given DR’s reliance on consensus. The authors also acknowledge the limitation to multiple-choice benchmarks. Overall, these issues relate to the main contribution/framing of the approach and are not easily resolved within a short rebuttal.

---

> > > ### Author Response · Authors · 2026-04-05
> > >
> > > Dear Reviewer Yjtb,
> > >
> > > We sincerely appreciate your continued engagement. We believe there may still be a misunderstanding regarding our paper's core motivation, and we would like to address them directly:
> > >
> > > (1) Our motivation does *not* actually involve potentially deceptive models; rather, it focuses on what a given protocol **incentivizes**. *DR doesn't just "hope" models are honest, but rather stops "forcing" them to lie.* On our assumption, just like other important works in this field (Khan et al., 2024; Irving et al., 2018), we assume a reasonable, baseline degree of reliable and cooperative capability from models.
> > >
> > > The core divergence is that **Debate's adversarial structure *creates* deceptive behavior by forcing models to defend fixed positions regardless of their internal beliefs (epistemic calibration), while DR aligns the protocol with models' genuine beliefs by allowing them to update and converge.** In Debate protocols, a model is structurally incentivized to consistently advocate for its assigned stance. This means that a model which internally realizes it is wrong is **structurally induced to act dishonestly**.
> > >
> > > Protocol design makes the critical difference here: DR removes this misalignment. Instead of forcing models into fixed advocacy roles that suppress their epistemic calibration (i.e., their ability to accurately weigh evidence, express appropriate uncertainty, and update their beliefs), DR leverages this intrinsic capability (which we empirically validate in Section 5.2). Therefore, DR is not relying on perfectly honest models as a naive assumption, but rather providing a protocol that does not actively induce them to be deceptive.
> > >
> > > (2) **Regarding agreement traps**, we fully acknowledge them as a meaningful failure mode for future research to look into. However, we do not leave this issue unaddressed: **both our theoretical (Theorem 3, Lemma 4) and empirical analyses (Section 5.2, Figure 2) demonstrate that improved consultant calibration mitigates this failure mode.** Furthermore, we explicitly trace these traps to specific symptoms of imperfect calibration, such as overthinking and stubbornness (Table 4), providing a concrete path for future mitigation. As one example of a potential detection mechanism: when consultants reach consensus but report low confidence, this could serve as a signal to flag likely agreement traps, our calibration analysis in Section 5.2 supports this, as it shows that well-calibrated models express higher confidence when they converge on the correct answer.
> > >
> > > ---
> > >
> > > Overall, DR outperforms Debate in 8 of 12 conditions with strong statistical significance, with the largest gains appearing precisely where scalable oversight is most needed: when the judge is weakest. While we acknowledge your remaining concerns regarding the limitations of our protocol, **we respectfully maintain that they do not invalidate our core contribution**. We believe the shift from adversarial interaction to collaborative resolution is both theoretically motivated and empirically supported, and that our empirical findings are undoubtedly useful for the field. **This constitutes a meaningful and novel step toward expanding the protocol design space beyond Debate, which remains the dominant paradigm in scalable oversight.**
> > >
> > > We greatly appreciate your feedback pointing out these concerns. It has helped us realize that our original writing and presentation likely contributed to these misunderstandings. To ensure these nuances are properly framed, we will update the draft by rewriting the introduction to contrast the systemic incentives of Debate vs. DR, and by adding a dedicated limitations subsection on the cause and mitigation of agreement traps. We sincerely hope you will reconsider your assessment and update the score in light of our efforts and these clarifications.

---

### Decision · Program_Chairs · 2026-04-30

**Decision:**

Accept (regular)

**Comment:**

The reviewers agree that this is an interesting and well-motivated approach to scalable oversight. In particular, they felt like the distinction between disagreement resolution and debate was meaningful and well-articulated. They also felt like the theoretical results were reasonable, despite resting on some rather strong assumptions. However, they collectively raised some points of concern, below:

1. MCQ datasets: One reviewer notes that using multiple-choice question datasets for evaluation does not match the potential real-world applications of such an approach, where presumably the answers will be open-ended and it will be difficult to determine which answer is "correct".

2. The agreement trap failure mode is novel and unique to DR. The authors provide some empirical and heuristic arguments about why this is not a fatal flaw, but it is an interesting observation that warrants further experimentation.

3. At the reviewers' request, the authors ran additional experiments during the rebuttal period. These would make the paper stronger, but it would be nice to have them fully peer-reviewed before adding them to the paper.